# Teprotumumab for the Treatment of Thyroid Eye Disease: Why Should We Keep Our Eyes “Wide Open”?—A Clinical and Pharmacovigilance Point of View

**DOI:** 10.3390/jpm14101027

**Published:** 2024-09-26

**Authors:** Arnaud Martel, Fanny Rocher, Alexandre Gerard

**Affiliations:** 1Ophthalmology Department, University Hospital of Nice, 06000 Nice, France; 2Pharmacovigilance Department, University Hospital of Nice, 06000 Nice, France; rocher.f@chu-nice.fr (F.R.); gerard.a@chu-nice.fr (A.G.)

**Keywords:** thyroid eye disease, teprotumumab, FDA approval, adverse drug reaction, cost, evidence-based medicine, conflicts of interest

## Abstract

**Objectives:** Thyroid eye disease (TED) treatment has been recently revolutionized with the approval of teprotumumab, a targeted insulin growth factor 1 receptor (IGF1R) inhibitor. To date, teprotumumab is the only FDA-approved drug for treating TED. In this article, we would like to temper the current enthusiasm around IGF1R inhibitors. **Methods:** critical review of the literature by independent academic practitioners. **Results:** several questions should be raised. First, “*how an orphan drug has become a blockbuster with annual sales exceeding $1 billion?*” Teprotumumab infusions are expensive, costing about USD 45,000 for one infusion and USD 360,000 for eight infusions in a 75 kg patient. Teprotumumab approval was based on two randomized clinical trials investigating active (clinical activity score ≥ 4) TED patients. Despite this, teprotumumab was approved by the FDA for “the treatment of TED” without distinguishing between active and inactive forms. The second question is as follows: “*how can a new drug, compared only to a placebo, become the new standard without being compared to historically established gold standard medical or surgical treatments?*” Teprotumumab has never been compared to other medical treatments in active TED nor to surgery in chronic TED. Up to 75% of patients may experience proptosis regression after treatment discontinuation. Finally, ototoxicity has emerged as a potentially devastating side effect requiring frequent monitoring. Investigation into the long-term side effects, especially in women of childbearing age, is also warranted. **Conclusions:** Teprotumumab is undoubtedly a major treatment option in TED. However, before prescribing a drug, practitioners should assess its benefit/risk ratio based on the following: (i) evidence-based medicine; (ii) their empirical experience; (iii) the cost/benefit analysis; (iv) the long-term outcomes and safety profile.

## 1. Introduction

Thyroid eye disease (TED) is an auto-immune disorder and represents the most common extrathyroidal manifestation of Grave’s disease [1]. TED can be divided into an active phase dominated by inflammatory symptoms followed by an inactive and chronic form [2]. According to the recommendations of the EUGOGO (European Group on Graves’ orbitopathy), intravenous steroids +/− associated with mycophenolate constitute the first-line treatment in active TED [3]. In case of failure, other options include more targeted therapies such as anti IL6 (tocilizumab) [4] or anti CD20 (rituximab) [5] drugs. Surgery remains the mainstay of treatment during the inactive phase and may comprise orbital decompression, strabismus surgery and eyelid surgery depending on patients’ symptoms [3]. 

TED treatment has been recently revolutionized with the approval of teprotumumab, a targeted insulin growth factor 1 receptor (IGF1R) inhibitor (IGF1Ri), by the Food and Drug Administration (FDA) in 2020 based on two clinical trials published in the prestigious *New England Journal of Medicine* in 2017 and 2020 [6,7]. Teprotumumab is currently the only FDA-approved drug for treating TED. It is widely prescribed in the United States (US) and is the focus of a growing number of publications. Teprotumumab is now considered the first-line treatment for moderate-to-severe TED with proptosis or diplopia by several Thyroid Societies [1]. 

TED pathophysiology has been extensively studied, and orbital fibroblasts, which carry IGF1R on their surface, appear to be a key therapeutic target [2]. By acting directly on orbital fibroblasts, IGF1Ri are currently the best targeted treatment available to reduce proptosis [7].

It is therefore legitimate to question whether teprotumumab is the long-awaited Holy Grail for treating TED [8]. In fact, several caveats should be highlighted to temper the current enthusiasm for IGF1Ri.

## 2. The FDA Approval

The FDA approval of teprotumumab is based on the results of two clinical trials in which only patients with active TED with a clinical activity score (CAS) ≥ 4 were included [6,7]. Teprotumumab was approved in 2020 for “the treatment of TED” without distinguishing between active and inactive forms. As a result, teprotumumab has been widely prescribed in the US in patients with both active and inactive TED, enabling the company marketing teprotumumab (Horizon Therapeutics) to collect retrospective data on its use in chronic TED (duration ≥ 2 years). Indeed, in a 2022 study conducted on 31 chronic TED patients who received at least three injections of teprotumumab, a mean reduction in proptosis of 3.5 mm was reported [9]. However, 18 patients (58%) had a CAS of 3, corresponding to active TED in Europe. This could have overestimated the actual effect of teprotumumab on the chronic phase of the disease. Based on this study, several Key Opinion Leaders (KOLs) have suggested that teprotumumab could replace orbital decompression in chronic TED. A 2023 prospective, randomized study on 62 chronic TED patients (42 on teprotumumab/20 on placebo) with a CAS ≤ 1 reported proptosis improvement by 2.41 mm and 0.9 mm in the teprotumumab and placebo groups, respectively (*p* = 0.004). It should be noted that proptosis regression was significantly lower in the prospective study (2.41 mm) than in the retrospective study (3.5 mm). Based on these findings, teprotumumab was granted an extension of approval by the FDA in April 2023 and is now officially indicated “for the treatment of TED regardless of TED activity or duration”.

## 3. Cost

Teprotumumab infusions are expensive, costing about USD 45,000 for one infusion and USD 360,000 for eight infusions in a 75 kg patient. In France, teprotumumab is available through an expanded access program, and treatment costs about EUR 450,000. Ongoing studies are investigating other types of IGF1Ri (NCT06021054) and their potential approval could reduce the cost of IGF1Ri. In addition, it should be noted that drug prices are generally lower in Europe compared to the United States. If teprotumumab is approved by the European Medicines Agency (EMA) in the coming weeks or months, the price of the product will be negotiated state by state. In France, the Economic Committee for Health Products is responsible for determining the exact reimbursement price of any medicine in the territory. National medico-economic assessments are warranted in the coming months to guide national economic committees.

## 4. Conflicts of Interest

TED estimated incidence ranges from 2.67 to 16/100,000 inhabitants/year [10]. Moderate-to-severe TED is even rarer, with an incidence of 0.05/100,000 inhabitants/year. TED is therefore considered an orphan disease in the US (with <200,000 Americans affected) and a rare disease in France (with an incidence < 1/2000 inhabitants/year). It is therefore surprising that an orphan drug has become a blockbuster (with annual sales exceeding USD 1 billion) in 2021 and 2022. At the end of 2023, Amgen acquired Horizon Therapeutics for USD 27.8 billion (it should be noted that Horizon Therapeutics was the owner of two other medications, pegloticase and inebilizumab-cdon, associated with other inflammatory disorders) [11]. Thus, how can such success be explained?

The first explanation is that teprotumumab has been widely prescribed since 2020 in the US.

In addition, many KOLs recruited as consultants for Horizon Therapeutics have published numerous articles and reported favorable results at conferences. In a personal communication at the 2023 European Society of Ophthalmic Plastic and Reconstructive Surgery (ESOPRS) meeting, our team highlighted that 68% of original clinical and/or scientific articles published in the literature on teprotumumab had as their first and/or last author someone who had been a consultant for Horizon Therapeutics. In an American Expert Consensus published in 2022 in the *Journal of Neuro-Ophthalmology*, 10/15 authors (66%) were consultants for Horizon Therapeutics, and one was the co-director of the laboratory [12]. 

Finally, it should be noted that companies manufacturing IGF1Ri (e.g., Horizon Therapeutics and Viridian Therapeutics) became major sponsors of European (ESOPRS) and American (ASOPRS) oculoplastics conferences in 2023.

## 5. Teprotumumab Shows the Limits of the Evidence-Based Medicine (EBM) Dogma

The success of teprotumumab raises a more fundamental question: how can a new drug, compared only to a placebo, become the new standard without being compared to historically established gold standard medical or surgical treatments? The answer is EBM. In 2022, a consensus statement developed by the American and European Thyroid Associations has recommended the use of teprotumumab as a first-line treatment in moderate-to-severe, active or progressive, TED associated with diplopia and/or proptosis, which corresponds to most patients encountered in clinical practice [1]. To date, six randomized placebo-controlled trials have been conducted in TED: three assessing teprotumumab [6,7,13], one assessing rituximab [14], one assessing tocilizumab [4], and one assessing intravenous steroids [15]. Mathematically, teprotumumab has the highest “level” of EBM, despite the lack of data on long-term outcomes and adverse drug reactions (ADRs). No randomized studies have compared teprotumumab to other medications. EBM has considerably improved the management of many diseases and is mainly based on randomized clinical trials. EBM has gradually supplanted empirical experience-based medicine [16]. TED is an autoimmune disorder, the course of which follows a predictable curve (Rundle’s curve) [17], although progressive TED without any inflammatory signs can also be encountered in daily clinical practice [18]. Only randomized trials can take into account the impact of the natural course on outcomes. In this context, three questions could be raised regarding the use of EBM in TED: (i) Is a new drug, the use of which is based on three recent randomized placebo-controlled trials, more effective than intravenous steroids that have been used successfully for almost a century, with a well-known safety profile, and which have previously been compared to other treatment options (oral steroids and radiotherapy)? (ii) Can a randomized placebo-controlled trial be considered sufficiently robust so that no comparative trial with other effective treatments is necessary? In other words, can a placebo-controlled trial be considered a “low level” of EBM? (iii) Steroids, tocilizumab and rituximab are widely prescribed in rheumatological and neurological diseases, which are much more prevalent than TED. The pharmaceutical companies marketing these drugs have little financial interest in conducting costly, large, prospective randomized studies in this indication. Could this influence the current level of evidence regarding their efficacy in TED?

## 6. Proptosis Recurrence

Proptosis regression is considered the main benefit of teprotumumab compared to other medications such as tocilizumab or rituximab. In the Optic X trial sponsored by Horizon Therapeutics, 26% of patients showed proptosis regression (≥2 mm) [13]. However, most studies have reported proptosis recurrence after teprotumumab discontinuation. Recently, a retrospective study has followed 78 patients treated with teprotumumab [19]. After a mean follow-up of 10.56 months, 59 patients (75%) experienced proptosis recurrence after treatment discontinuation. Another retrospective study reported that of 21 active TED patients followed more than 6 months after stopping teprotumumab, approximately two thirds had regression of CAS and/or proptosis (≥2 mm), with average regression occurring at 12.25 months [20]. Another retrospective study attempted to compare teprotumumab (n = 51; mean follow-up: 8.2 months) to orbital decompression (n = 77; mean follow-up 14.4 months) and found that patients treated with teprotumumab had lower slight increase of 0.68 mm in proptosis, while patients treated with orbital decompression showed a slight decrease in proptosis of 0.18 mm [21]. These studies highlight the need for long-term data.

## 7. Adverse Drug Reactions

IGF1R is expressed in the inner ear. The prevalence of ototoxicity in patients treated with teprotumumab is difficult to assess due to conflicting data. Industry-sponsored studies have reported an ototoxicity rate of 10% [7,22], while other studies have reported a higher prevalence (up to 90% based on audiometry recordings) [23]. Several cases of irreversible deafness have been reported, possibly due to preexisting hearing disorders. A 2024 study assessed hearing loss in patients treated with teprotumumab compared to other patients based on the FDA Adverse Event Reporting System [24]. Teprotumumab was associated with a 24-fold increased risk of hearing problems, including deafness, eustachian tube disorders and tinnitus. In addition, the FDA added an updated warning in July 2023 about teprotumumab-related hearing damage. Therefore, audiometric monitoring is now recommended before, during and after teprotumumab administration. In the World Health Organization global adverse event reports database (VigiBase^®^), 3271 teprotumumab-related ADRs have been identified, including 1146 (35%) and 22 (0.7%) serious and fatal ADRs, respectively. More specifically, 750 (23%) ADRs concerned hearing disorders, including 153 (4.7%) cases of deafness and 72 (2.2%) cases of irreversible deafness. 

Pharmacovigilance and long-term follow-up are also mandatory in women of childbearing age, as menstrual irregularities and amenorrhea have been reported with teprotumumab [25].

## 8. Looking to the Future

Based on this critical Editorial, we recommend that the following points be addressed in the coming years:(i)State-by-state medical–economic assessment of the impact of IGF1-R antagonists compared to other treatment strategies, including but not limited to IV steroids, anti-IL6 and orbital decompression, should be carried out. Future medical–economic directions could include full reimbursement, conditional reimbursement (in case of failure of previous, less costly treatments), and no reimbursement at all.(ii)Comparative studies not against a placebo but against other historical and more recent and promising drugs are justified. Until now, anti-IGF1-R drugs were only compared to a placebo, thus artificially increasing their results. Direct comparisons with IV steroids, anti-IL6 drugs, and even orbital decompression are urgently needed.(iii)Will the pharmaceutical industry be inclined to conduct expensive therapeutic trials against historical treatments with the risk of obtaining less impressive (or even worse) results? We personally have some doubts. In our opinion, the only way to overcome this problem is to conduct an industry-independent, prospective, randomized, comparative trial under national supervision. The main limitation remains the price of such a trial. For 30 years in France, the National Hospital Clinical Research Program (PHRC) has made it possible to independently conduct such large-scale trials. We are currently studying a project under the supervision of the French government.

## 9. Conclusions

Teprotumumab is undoubtedly a major treatment option in TED. However, before prescribing a drug, practitioners should assess its benefit/risk ratio based on the following: (i) the EBM; (ii) the empirical experience; (iii) the cost/benefit analysis; (iv) the long-term outcomes and safety profile. It must be remembered that a good treatment is the right drug prescribed for the right indication. Although teprotumumab is highly effective in active TED, its superiority to other less expensive drugs available has not been demonstrated. Its place in chronic TED management should thus be reevaluated. At the time this Editorial was written, the pharmaceutical company marketing teprotumumab announced the imminent submission of a marketing authorization application to the European Medicines Agency.

## Data Availability

Not applicable.

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
