# Peer review of "Teprotumumab for the Treatment of Thyroid Eye Disease: Why Should We Keep Our Eyes “Wide Open”?—A Clinical and Pharmacovigilance Point of View"

_jpm, 2024, doi:10.3390/jpm14101027_

Round 1

Reviewer 1 Report

Comments and Suggestions for Authors

line 83-84 delete the sentence too speculstive

line 114-15 Although the Rundle curve is used to describe the course of TED ,this is only of historical interest and many cases do not follow the predictable Rundle curve

Comments on the Quality of English Language

Line 48 instead of warnings consider other words like caveats

Line 53 delete “surprisingly” 

line 141 instead of “ industrial “ use industry-sponsored

line 164 - remove the word “remember that “

line 167 - instead of tampered consider “ reevaluated”

Author Response

On behalf of all authors of the study, I would like to thank all reviewers for their valuable work, contribution and interesting comments. We are very grateful. Please find our point-by-point answers below. Changes are highlighted in yellow in the revised manuscript.

line 83-84 delete the sentence too speculstive

Thank you for your remark. However, the following sentence “At the end of 2023, Amgen has acquired Horizon Therapeutics for $27.8 billion. Thus, how can such success be explained?” line 83-84 is not speculative but 100% factual. This can be found in the AMGEN website (https://www.amgen.com/newsroom/press-releases/2023/10/amgen-completes-acquisition-of-horizon-therapeutics-plc). We have added a reference accordingly. To be fully honest, it should be noted that Horizon Therapeutics was the owner of 2 other medications: pegloticase and inebilizumab-cdon indicated in other inflammatory disorders. This has been added in the revised manuscript.

This is a very important point of the article which highlights the financial issues posed by the development of anti-IGF1R drugs. It would be too speculative, so we will not do so, to state that most IGF1R inhibitor manufacturers are conducting clinical trials against placebo and not against historical treatments to obtain better results with the aim of being acquired by larger pharmaceutical industries, as happened with the Horizon Therapeutics story.

line 114-15 Although the Rundle curve is used to describe the course of TED ,this is only of historical interest and many cases do not follow the predictable Rundle curve

We agree with you. We have added the reference of the recently published Mayo clinical article in OPRS on the “Quiet orbitopathy”. The following sentence has been added in the revised manuscript: “TED is an autoimmune disorder, the course of which follows a predictable curve (Rundle’s curve) [15] although progressive TED without any inflammatory signs can also be encountered in daily clinical practice [16]”

Comments on the Quality of English Language

Line 48 instead of warnings consider other words like caveats

Thank you for your comment. We have used the term caveats accordingly.

Line 53 delete “surprisingly”

This has been deleted accordingly.

line 141 instead of “ industrial “ use industry-sponsored

We have replaced industrial with industry-sponsored accordingly

line 164 - remove the word “remember that “

This word has been removed accordingly

line 167 - instead of tampered consider “ reevaluated”

we have replaced tempered with reevaluated accordingly

Thank you for your comments.

Reviewer 2 Report

Comments and Suggestions for Authors

The opinion written by Martel and colleagues addresses various aspects of the introduction to therapy, costs, and limitations associated with the use of teprotuzumab, a targeted insulin growth factor 1 receptor (IGF1R) inhibitor, in the treatment of Graves’ orbitopathy. The article is well-written and provides interesting insights into introducing new therapies into treatment that we may often not be aware of.

Author Response

On behalf of all authors of the study, I would like to thank all reviewers for their valuable work, contribution and interesting comments. We are very grateful. Please find our point-by-point answers below. Changes are highlighted in yellow in the revised manuscript.

The opinion written by Martel and colleagues addresses various aspects of the introduction to therapy, costs, and limitations associated with the use of teprotuzumab, a targeted insulin growth factor 1 receptor (IGF1R) inhibitor, in the treatment of Graves’ orbitopathy. The article is well-written and provides interesting insights into introducing new therapies into treatment that we may often not be aware of.

We would like to thank you for your comment.

Reviewer 3 Report

Comments and Suggestions for Authors

This opinion provides a balanced view by comparing results from retrospective and prospective studies, on teprotumumab and its effects on thyroid eye diseases.

Here are my comments :

- I encourage the authors to add the type of the manuscript in the title, to prevent it from being considered as a review in the beginning

- Introduction could be better enhanced by adding more background on the existing data of other drugs of this disease , before talking about teprotumumab 

- The discussion on FDA approval is critical and well-placed. I highly encourage the authors to maintain a neutral usage of words in their titles : change ' mystery ' in the title and replace it by a neutral term.

because the observed reduction in proptosis was observed in some studies, however there is a need for further research to confirm the drug's effectiveness across different stages of TED.

- The discussion on cost is highly relevant, given the significant expense associated with teprotumumab. Are there potential cost-effective alternatives or ways to negotiate drug prices that could improve accessibility?

- The authors must add their opinion on what further research is needed to establish the drug's efficacy in chronic TED cases?

- The authors must include if there are trials that might provide more clarity on its use in different TED stages

Comments on the Quality of English Language

minor editing

Author Response

On behalf of all authors of the study, I would like to thank all reviewers for their valuable work, contribution and interesting comments. We are very grateful. Please find our point-by-point answers below. Changes are highlighted in yellow in the revised manuscript.

This opinion provides a balanced view by comparing results from retrospective and prospective studies, on teprotumumab and its effects on thyroid eye diseases.

Here are my comments :

- I encourage the authors to add the type of the manuscript in the title, to prevent it from being considered as a review in the beginning

We agree with you and modified the title accordingly: “Teprotumumab for the treatment of thyroid eye disease: why should we keep our eyes “wide open”? A clinical and pharmacovigilance point of view”

- Introduction could be better enhanced by adding more background on the existing data of other drugs of this disease , before talking about teprotumumab

We agree with you. We have added the following paragraph in the introduction.

Thyroid eye disease (TED) is an auto-immune disorder and represent the most common extrathyroidal manifestation of Grave’s disease [1]. TED can be divided into an active phase dominated by inflammatory symptoms followed by an inactive and chronic form [2]. According to the recommendations of the  EUGOGO (European Group on Graves' orbitopathy), intravenous steroids +/- associated with mycophenolate constitute the first-line treatment in active TED [3]. In case of failure, other options include more targeted therapies such as anti IL6 (tocilizumab) [4] or anti CD20 (rituximab) [5] drugs. Surgery remains the mainstay of treatment during the inactive phase and may comprise orbital decompression, strabismus surgery and eyelid surgery depending on patients’ symptoms[3].

- The discussion on FDA approval is critical and well-placed. I highly encourage the authors to maintain a neutral usage of words in their titles : change ' mystery ' in the title and replace it by a neutral term.

Although we believe the nature of the FDA approval remains a mystery (approved for active AND INACTIVE disease), we have removed the term mystery to improve the neutrality of the title.

because the observed reduction in proptosis was observed in some studies, however there is a need for further research to confirm the drug's effectiveness across different stages of TED.

- The discussion on cost is highly relevant, given the significant expense associated with teprotumumab. Are there potential cost-effective alternatives or ways to negotiate drug prices that could improve accessibility?

Thank you for your very interesting comment. We have added the following informations in the cost paragraph. Taken together, we believe that competition (from other IGF1R inhibitor companies) and state-by-state negotiation in the European Union (in case of approval by the EMA) will drive the price of the product down, although it is likely that the final price will be higher than the IV steroids-surgery recommendation provided by the EUGOGO.

Modifications in the manuscript : In addition, it should be noted that drug prices are generally lower in Europe compared to the United States. If teprotumumab is approved by the European Medicines Agency (EMA) in the coming weeks or months, the price of the product will be negotiated state by state. In France, the Economic Committee for Health Products is responsible for determining the exact reimbursement price of any medicine in the territory. National medico-economic assessments are warranted in the coming months to guide national economic committees.

- The authors must add their opinion on what further research is needed to establish the drug's efficacy in chronic TED cases? The authors must include if there are trials that might provide more clarity on its use in different TED stages

Thank you for this remark. Initially we did not want to provide any personal recommandations. We have added a nex paragraph entitled “And next?”

And next?

Based on this critical editorial, we recommend that the following points be addressed in the coming years:

(i)           State-by-state medical-economic assessment of the impact of IGF1-R antagonists compared to other treatment strategies, including but not limited to IV steroids, anti-IL6 and orbital decompression. Future medical-economic directions could include full reimbursement, conditional reimbursement (in case of failure of previous, less costly treatments), and no reimbursement at all.

(ii)          Comparative studies not against a placebo but against other historical and more recent and promising drugs are justified. Until now, anti-IGF1-R drugs were only compared to a placebo, thus artificially increasing their results. Direct comparisons with IV steroids, anti-IL6 drugs, and even orbital decompression are urgently needed.

(iii)         Will the pharmaceutical industry be inclined to conduct expensive therapeutic trials against historical treatments with the risk of obtaining less impressive (or even worse) results? We personally have some doubts. In our opinion, the only way to overcome this problem is to conduct an industry-independent, prospective, randomized, comparative trial under national supervision. The main limitation remains the price of such a trial. For 30 years in France, the National Hospital Clinical Research Program (PHRC) has made it possible to independently conduct such large-scale trials. we are currently studying a project under the supervision of the French government.

Reviewer 4 Report

Comments and Suggestions for Authors

Thank you for the opportunity to review the opinion entitled "Teprotumumab for the treatment of Thyroid Eye Disease: why should we keep our eyes “wide open”?"

I read the article with great attention, because it deals with an important topic, which is the effective treatment of Thyroid Eye Disease - the article presented to me for review is an opinion of 3 authors about teprotumumab.

The "Introduction" section briefly introduces the reader to the further elements of the manuscript. I have no objections to it.

The title of the next section "The mystery surrounding FDA approval" seems a bit speculative. While it may aim to engage the reader, it does not accurately represent the actual circumstances. I recommend revising the title to something like "FDA approval and associated doubts" or something like that.

The next section "Cost" is a very short section (4 lines), to which the authors did not add any reference. It also lacks information on social costs and, for example, real annual expenses for treatment with teprotumumab. This information would be worth supplementing.

The section "Conflicts of Interest" does not raise any objections.

The content of the section "Teprotumumab shows the limits of the Evidence-Based Medicine (EBM) dogma" is interesting, although I also consider the title to be somewhat speculative. The arguments presented by the authors do not show the EBM doctrine's weakness, but rather its understanding in the scientific world. I would consider changing the title, although leaving it in this form does not disqualify the manuscript as a whole.

The amount of data in the section "Proptosis recurrence" is quite sparse. If the authors describe that "most studies have reported proptosis recurrence after teprotumumab discontinuation", then they should present hard data on disease recurrence. Only two papers received citations, despite a larger number of studies in the existing literature. There is a clear need to expand and supplement this section with additional data.

The section "Adverse drug reactions" raises no objections - as does "Conclusions".

I suggest a major revision of this article.

Author Response

On behalf of all authors of the study, I would like to thank all reviewers for their valuable work, contribution and interesting comments. We are very grateful. Please find our point-by-point answers below. Changes are highlighted in yellow in the revised manuscript.

 Thank you for the opportunity to review the opinion entitled "Teprotumumab for the treatment of Thyroid Eye Disease: why should we keep our eyes “wide open”?"

I read the article with great attention, because it deals with an important topic, which is the effective treatment of Thyroid Eye Disease - the article presented to me for review is an opinion of 3 authors about teprotumumab.

The "Introduction" section briefly introduces the reader to the further elements of the manuscript. I have no objections to it.

The title of the next section "The mystery surrounding FDA approval" seems a bit speculative. While it may aim to engage the reader, it does not accurately represent the actual circumstances. I recommend revising the title to something like "FDA approval and associated doubts" or something like that.

Thank you for your comment which was shared by another reviewer. The title seemed too speculative. Therefore, we have changed the title and reduced it to: “FDA Approval”.

The next section "Cost" is a very short section (4 lines), to which the authors did not add any reference. It also lacks information on social costs and, for example, real annual expenses for treatment with teprotumumab. This information would be worth supplementing.

Thank you for your valuable comment. We have modified the paragraph accordingly.

Modifications: In addition, it should be noted that drug prices are generally lower in Europe compared to the United States. If teprotumumab is approved by the European Medicines Agency (EMA) in the coming weeks or months, the price of the product will be negotiated state by state. In France, the Economic Committee for Health Products is responsible for determining the exact reimbursement price of any medicine in the territory. National medico-economic assessments are warranted in the coming months to guide national economic committees.

Of note, it should be noted that current medical-economic studies comparing teprotumumab with historical treatments (IV steroids and orbital decompression according the EUGOGO guidelines for active and chronic phases, respectively) are currently lacking.  In a new paragraph entitled “what next?” we recommend state-by-state medical-economic assessments in Europe (if approved by the EMA, which it should be).

The section "Conflicts of Interest" does not raise any objections.

Thank you

The content of the section "Teprotumumab shows the limits of the Evidence-Based Medicine (EBM) dogma" is interesting, although I also consider the title to be somewhat speculative. The arguments presented by the authors do not show the EBM doctrine's weakness, but rather its understanding in the scientific world. I would consider changing the title, although leaving it in this form does not disqualify the manuscript as a whole.

We completely understand your comment. For 30 years, EBM has become the gold standard and the randomized clinical trial the Holy Grail. Today, only randomized clinical trials are taken into account to establish national and international guidelines, as demonstrated by the American and European thyroid associations in TED. EBM is important in TED because of its natural course and the possibility of spontaneous improvement. EBM is even more essential in oncology. Let's focus on tebentafusp in metastatic uveal melanoma. Two articles were published in the New England Journal of Medicine, as were the TED articles by Smith and Douglass. What is the main difference between tebentafusp articles and TED articles? The control group. In the articles on tebentafusp, the control group is not a placebo but the current historical treatments (anti PD1 or dacarbazine) while in the TED articles, the IGF1R inhibitors are compared only to the placebo and not to the historical treatments. Although both articles are well-conducted multicenter randomized trials, the level of scientific evidence is not identical in our opinion. However, these differences in terms of control group are not sufficiently highlighted in the current literature. Of course, administering a placebo would not be ethical in oncology. But is it ethical to administer a placebo to a TED patient with a CAS score of 7/7? We therefore wish to keep our title as presented in the initial version of our article

The amount of data in the section "Proptosis recurrence" is quite sparse. If the authors describe that "most studies have reported proptosis recurrence after teprotumumab discontinuation", then they should present hard data on disease recurrence. Only two papers received citations, despite a larger number of studies in the existing literature. There is a clear need to expand and supplement this section with additional data.

Thank you for your valuable remark. Long term data regarding proptosis recurrence are quite square. Most studies report results at the end of the last infusion or few weeks thereafter. We have added several small sample size studies which have investigated long-term proptosis recurrence after teprotumumab discontinuation.

Modifications: Another retrospective study reported that of 21 active TED patients followed more than 6 months after stopping teprotumumab, approximately two thirds had regression of CAS and/or proptosis (≥2 mm), with average regression occurring at 12.25 months [20]. Another retrospective study attempted to compare teprotumumab (n = 51; mean follow-up: 8.2 months) to orbital decompression (n = 77; mean follow-up 14.4 months) and found that patients treated with teprotumumab had lower slight increase of 0.68 mm in proptosis while patients treated with orbital decompression showed a slight decrease in proptosis of 0.18 mm [21].These studies highlight the need for long-term data.

The section "Adverse drug reactions" raises no objections - as does "Conclusions".

Thank you

Round 2

Reviewer 4 Report

Comments and Suggestions for Authors

Thank you for the opportunity to do a second review of the manuscript titled "Teprotumumab for the Treatment of Thyroid Eye Disease: Why Should We Keep Our Eyes 'Wide Open'?"

The authors adequately dealt with the prior comments, and the revisions to the manuscript have significantly enhanced its quality.

I have no more suggestions.

I suggest accepting the manuscript for publication in its current format.